# Exploring the socioeconomic determinants of dietary diversity in rural Bangladesh: A longitudinal study

**Tayaba Cheragee Prachee**[1☺], **Md Rasel Biswas**[1☺*], **Saiful Islam**[2]

1 Institute of Statistical Research and Training, University of Dhaka, Dhaka, Bangladesh, 2 Institute of Nutrition and Food Sciences, University of Dhaka, Dhaka, Bangladesh

☺ These authors contributed equally to this work.

* rasel@du.ac.bd

**Data availability statement:** All three data files are available from the Harvard Dataverse

## Abstract

Despite significant progress in food and nutrition security over the past few decades, Bangladesh continues to face challenges, especially in rural areas. This study investigates the relationship between socioeconomic determinants and household dietary diversity using three waves of nationally representative longitudinal data from the Bangladesh Integrated Household Survey (2011/2012, 2015, and 2018/2019). Dietary diversity was measured using the Household Dietary Diversity Score (HDDS) and the Food Variety Score (FVS), both indicators of dietary diversity and food security. We employed a Poisson fixed-effects regression model with robust standard errors to model the dietary diversity indicators, appropriate for the count nature of the data. The results show that households where the head is engaged in farming have, on average, a 1.4% higher HDDS and a 2.17% higher FVS compared to non-farming households. Households where the head has primary or lower education demonstrate a 2.11% increase in HDDS, while those with secondary or higher education experience a 2.05% increase compared to households with no formal education. Additionally, each extra household member increases FVS by 5.5%. Cultivable land, as well as access to essential resources like water and electricity, significantly affects dietary diversity. Economic factors, such as higher food and non-food expenditures per capita, also exhibit strong positive associations with dietary diversity. These findings emphasize the importance of socioeconomic factors and household dynamics in shaping dietary quality. Policy interventions should prioritize educational programs to enhance nutrition knowledge and promote sustainable agricultural practices to support farmers. Aligning with the Sustainable Development Goals (SDGs), particularly SDG 2 on zero hunger and SDG 3 on health and well-being, these insights contribute to ongoing efforts to improve food security and dietary diversity in rural communities.

## Introduction

Food security is defined as a situation that exists "when all people, at all times, have physical, social and economic access to sufficient, safe and nutritious food that meets their dietary

database (https://doi.org/10.7910/
DVN/OR6MHT, https://doi.org/10.7910/
DVN/BXSYEL, and
https://doi.org/10.7910/DVN/NXKLZJ).

**Funding:** The author(s) received no specific
funding for this work.

needs and food preferences for an active and healthy life"[1]. Despite ongoing efforts, food insecurity and malnutrition remain critical global challenges, making food a core component of the Sustainable Development Goals (SDGs). The second of the UN's 17 SDGs is to "End hunger, achieve food security and improved nutrition and promote sustainable agriculture" by 2030 [2].

Food security is typically conceptualized through three essential pillars: food availability, food access, and food utilization [3,4]. Previous research has shown that dietary diversity plays a vital role in all three pillars [5–7]. Dietary diversity refers to the variety or number of distinct food groups—such as vitamins, minerals, proteins, and fats—consumed over a certain period [8,9].

Several indicators have been used to assess dietary adequacy, with the most common being the Household Dietary Diversity Score (HDDS) and the Food Variety Score (FVS) [7,10,11]. The HDDS measures the range of food items consumed by a household over a specified period, providing insights into both nutritional adequacy and access to diverse foods. In contrast, the FVS provides a deeper look into the dietary habits of household members by evaluating all types of food consumed, whether prepared at home or sourced from elsewhere, over a given recall period. Together, these indicators provide a comprehensive picture of both the quality and richness of a household's diet, shedding light on key aspects of food security and well-being.

Using data from ten low- and middle-income countries, including Bangladesh, Hoddinott and Yohannes [12] found a significant association between dietary diversity and food access. Similarly, Bidisha et al. [13] employed the HDDS to explore the relationship between access to credit and dietary diversity in Bangladesh. In Malaysia, Ihab et al. [14] investigated the relationship between food security, dietary diversity, and food expenditure, revealing that higher food expenditure correlates with increased dietary diversity, highlighting the importance of economic factors in shaping dietary habits. Khandoker et al. **(author?)**[15] examined the link between farm production diversity and dietary diversity using both the HDDS and FVS. Their study found that farm production diversity, income, and dietary diversity were positively associated. Additionally, factors such as market access, age, and education were identified as significant determinants of household dietary diversity. Dietary diversity is a pressing concern in Bangladesh, particularly in rural areas, where it is notably lower than in urban regions. While earlier studies have established a link between dietary diversity and agricultural diversification, the influence of socioeconomic factors—such as the gender, occupation, and education of the household head, as well as expenditure patterns and access to resources—on household dietary diversity in rural Bangladesh remains underexplored. Most prior research has relied on cross-sectional data, limiting insights into the temporal dynamics of dietary diversity. Variations in food access over the years can also influence dietary patterns, as events like natural disasters or economic shocks may disrupt food availability, leading to decreased dietary diversity.

To address these gaps, this study uses balanced panel data from the Bangladesh Integrated Household Survey (BIHS) conducted in 2011/2012, 2015, and 2018/2019 to analyze household dietary diversity in rural Bangladesh. The use of panel data allows us to track the same households over multiple time periods, which helps account for both temporal changes and household-level dynamics in dietary diversity. This approach improves the robustness of our findings by controlling for unobserved household-specific factors that may remain constant over time, thereby reducing bias from omitted variables. We employ a Poisson regression model, which is particularly appropriate for this analysis because both the HDDS and the FVS are count variables representing the number of different food groups consumed. The Poisson model is well-suited for handling non-negative integer data and allows us to model the

probability of observing different counts of dietary diversity. Additionally, incorporating fixed effects in the Poisson model helps control for unobserved heterogeneity among households. This combined approach effectively captures the dynamic nature of food access and dietary patterns over time while addressing potential overdispersion using robust standard errors.

## Data

This study uses data from three rounds of the Bangladesh Integrated Household Survey (BIHS) conducted in 2011/2012, 2015, and 2018/2019, focusing exclusively on rural areas of Bangladesh [16–18]. The BIHS is distinguished from other national surveys, such as the Household Income and Expenditure Survey (HIES), due to its comprehensive data collection methodology. In addition to collecting data on variables similar to those found in the HIES, the BIHS differentiates itself by addressing specific dimensions such as agricultural practices, dietary habits for a comprehensive understanding of eating patterns, various health metrics, and women's empowerment as assessed through the Women's Empowerment in Agriculture Index (WEAI). The survey covers rural areas in all seven administrative divisions of Bangladesh: Barisal, Chittagong, Dhaka, Khulna, Rajshahi, Rangpur, and Sylhet; it also includes rural areas in the *Feed the Future (FTF)* Zone of Influence in southwest Bangladesh. This dataset serves as a valuable resource for policymakers, researchers, and organizations addressing diverse socioeconomic challenges in the country. The BIHS was designed and implemented by the International Food Policy Research Institute (IFPRI), and the terms, conditions, and data usage guidelines of the IFPRI were strictly followed in conducting this study.

## Sampling

The BIHS sample was collected using a stratified sampling design with two stages: (1) selection of primary sampling units (PSUs) and (2) selection of households within each PSU. total of 325 PSUs were selected from 8 strata—the 7 administrative divisions and the FTF zone— using probability proportional to size (PPS) sampling. From each PSU, 20 households were randomly chosen, resulting in a sample of 6,503 households in the initial survey. The classification at the division level for the 6,503 households in 2012, 2015, and 2018 reveals that 4,423, 4,619, and 4,886 households, respectively, are designated as "Nationally Representative", representing rural Bangladesh. The expansion of this survey involved the addition of 52 PSUs, contributing 1,040 sample households, selected from FTF upazilas in Barisal, Dhaka, and Khulna strata (divisions) within the overall BIHS sampling framework. This study includes observations from the seven administrative division strata, excluding those initially selected from the FTF stratum. We constructed a balanced panel by tracking the same households over three rounds. Households with missing data on the sex of the head or food intake, even in a single round, were excluded. Additionally, households that split into separate units during the study period were excluded. The final panel dataset consists of 4,593 households with complete information across all three rounds. (see Table 1).

## Variables

This study investigates the relationship between household dietary diversity and various socioeconomic characteristics of the household head, along with other household attributes. Data collected includes dietary diversity measures, as well as socioeconomic factors such as the age, education, and occupation of the household head, and additional variables like household size and expenditure. Several studies have used the Household Dietary Diversity

**Table 1. Description of the Panel Data.**

| Items | Round 1 (2012) | Round 2 (2015) | Round 3 (2018) | Total | Balanced Panel |
|---|---|---|---|---|---|
| Number of households in BIHS data | 6503 | 6437 | 5605 | 18545 | |
| Number of households selected | 4,593 | 4,593 | 4,593 | 13,779 | 4,593 |

Score (HDDS) and the Food Variety Score (FVS) to assess dietary diversity [11,19]. In this study, HDDS and FVS serve as the outcome variables, enabling an in-depth examination of how socioeconomic factors and household characteristics influence dietary practices.

## Measurement of variables

1. **Household Dietary Diversity Score (HDDS)**
   Type: Count variable.
   Description: This study evaluates the HDDS using a 7-day recall period from the BIHS data. The HDDS ranges from 0 to 12, reflecting the number of distinct food groups consumed by a household.
   Calculation: The HDDS is calculated as the sum of 12 predefined food groups consumed by the household (see Table 2). The HDDS indicator is specifically designed and validated to measure household-level access to food, rather than individual-level access [20]. In rural, low-income settings, it is uncommon for household members to purchase or consume food outside the home, as meals are generally prepared and consumed within the household, even by those working away during the day. Therefore, foods consumed outside the home and not prepared by the household are excluded from the HDDS calculation. The HDDS is calculated as:
   The HDDS is calculated as

$$HDDS = F1 + F2 + \cdots + F12,$$

   where $F1, F2, \ldots, F12$ are binary variables representing whether a household consumed the corresponding food group (1 if consumed, 0 otherwise).

2. **Food Variety Score (FVS)**
   Type: Count variable.

**Table 2. The Food Groups Used for Calculating Household Dietary Diversity Score (HDDS).**

| Notation | Food groups |
|---|---|
| F1 | Cereals, rice, wheat |
| F2 | Roots and tubers |
| F3 | Vegetables, leafy vegetables |
| F4 | Fruits |
| F5 | Meat |
| F6 | Eggs |
| F7 | Fish (large/small) and seafood |
| F8 | Pulses, legumes, nuts |
| F9 | Milk and milk products |
| F10 | Oil/fats |
| F11 | Sugar/honey |
| F12 | Miscellaneous (spices, drinks, and beverages) |

Description: The FVS provides a detailed count of the individual foods, mixtures, and combinations consumed by household members. This measure offers insights into the participants' nutritional choices and meal patterns.

Calculation: The FVS is calculated by counting how many different food groups were consumed at least once over a period of 7 days [21]. These food groups can include both single food items, such as meat, fish, or vegetables, as well as mixed foods, like prepared dishes that combine multiple ingredients, such as khichuri or cooked firni. Importantly, this includes food prepared both inside the home and outside, such as from restaurants or street vendors. The goal of the FVS is to assess the diversity of the diet based on the number of different food groups consumed.

This study examines household dietary diversity at the aggregate level rather than the individual level. Accordingly, gender is analyzed in the context of the household, specifically through the gender of the household head (1 for male, 0 for female). This approach aligns with the study's household-level focus and enables an assessment of the broader influence of gender on household dietary diversity. The other independent variables used in this study, based on existing research, include the following characteristics of the household head: age, religion (1 for Islam, 0 for other), educational attainment (0 for no education, 1 for primary or lower level education, 2 for secondary or higher level education), main occupation (1 for farming, 0 for other occupations), marital status (1 for married, 0 for otherwise), and household size. As a substitute for income, we consider expenditure to be another explanatory variable. Since direct expenditure data was not available for households in our dataset, we are using household food consumption expenditure per capita (sum of expenses on purchased food consumed by the household in the last 7 days, excluding the value of food received as gifts, divided by household size to yield a per capita measure), and household non-food consumption expenditure per capita (sum of monthly expenses on non-food items such as fuel, lighting, cosmetics, cleaning supplies, transport, and other miscellaneous charges, divided by household size to provide a per capita assessment of non-food expenditures). Regarding wealth, while we did not explicitly include total wealth as a covariate, it is important to recognize that in rural contexts, common wealth indicators include cultivable land or pond (in decimal), ownership and home ownership (1 for owned, 0 for otherwise). Both of these serve as practical proxies for wealth. Land ownership reflects access to productive assets, while home ownership reflects financial stability and long-term investments. Therefore, the variables, income and wealth, are considered indirectly through related measures in our analysis. Furthermore, variables measuring household access to electricity and water supply, and the experience of a negative shock (1 for experienced, 0 for not experienced) were also included. The divisions are also considered as geographical locations in this study, as they provide a contextual framework for analyzing household dietary diversity across different regions. By categorizing the data based on these divisions—Barisal, Khulna, Dhaka, Chittagong, Rajshahi, Rangpur, and Sylhet—the study accounts for potential regional variations that may influence dietary patterns and diversity at the household level.

## Methodology

### Exploratory analysis

This study provides a comprehensive overview of the variables through the exploratory data analysis techniques. For continuous variables of interest, the mean and standard deviation are presented across the three survey rounds, offering insights into central tendency and variability. For categorical variables, frequency and percentage distributions are reported, allowing

an examination of their patterns over time. Furthermore, to assess whether significant differences exist between survey-round means, adjusted F-statistics were employed [22,23]. This is a generalization of the traditional F-test, modified to appropriately account for survey design features, including clustering, stratification, and unequal weighting.

## Poisson regression model

In this study, we employed Poisson regression to model household dietary diversity, as the outcome variables (HDDS and FVS) are counts. Count data are non-negative integers and often skewed, violating fundamental assumptions of linear regression, such as normality and homoscedasticity. Although transformations (e.g., logarithmic or square root) might allow for the application of linear regression, they complicate the interpretation of results and fail to accommodate the inherently discrete nature of the data. Poisson regression is ideally suited for this analysis as it directly models count outcomes using a log-link function and assumes a Poisson distribution for the response variables.

Let $DD_{it}$ denote the dietary diversity indicator (either HDDS or FVS) for household $i$ at survey round $t$. A panel Poisson regression model with $k$ dimensional vector of explanatory variables $\mathbf{X}_{it}$ can be specified as

$$\ln(\lambda_{it}) = \beta'_X \mathbf{X}_{it} + \mathbf{C}_i + \mathbf{S}_t,$$

with $DD_{it}$ following the Poisson distribution

$$P\left(DD_{it}\right) = \frac{e^{-\lambda_{it}} \lambda_{it}^{DD_{it}}}{DD_{it}!}, \quad \text{and} \quad E[DD_{it} \mid \mathbf{X}_{it}] = \lambda_{it},$$

where $\lambda_{it}$ is the conditional mean outcome for household $i$ at round $t$, $\mathbf{X}_{it}$ are the vectors of explanatory variables, $\beta_X$ is the vector of coefficients associated with $\mathbf{X}_{it}$, and $\mathbf{C}_i$ and $\mathbf{S}_t$ are the unobserved household-specific and round-specific effects. For detailed information on the Poisson regression model, see [24].

To determine whether a fixed-effects or random-effects model is more appropriate, the Hausman test was conducted for both outcome variables. The results indicate that the fixed-effects Poisson model is better suited for these variables. This result aligns with prior studies, where fixed-effects Poisson regression models have been used for analyzing panel data on dietary diversity [11,25] and pesticide exposure [26], ensuring control for unobserved heterogeneity. To address the unobserved heterogeneity and intra-household correlation, the study uses a robust estimator in place of the asymptotic covariance matrix. This approach adjusts for violations of model assumptions and accounts for overdispersion by employing a pseudo-likelihood function rather than the true log-likelihood function [27]. Given the nonlinearity of the Poisson model, the coefficients are interpreted as changes in the rate on a logarithmic scale in response to changes in the explanatory variables. The results are presented in terms of the Incidence Rate Ratio (IRR), providing a clearer interpretation of how a one-unit change in an independent variable affects the outcome.

## Software used

The analyses for this study were conducted using Stata 17 [28]. To account for the complex sampling design, the `svyset` command was utilized, ensuring the application of appropriate sampling weights and the identification of cluster and stratum variables. The `svy:` command was further employed to ensure proper analysis of survey data.

### Ethical considerations

This study utilized publicly available data from three rounds of the Bangladesh Integrated Household Survey (BIHS), conducted in 2011/2012, 2015, and 2018/2019. The BIHS was designed and implemented by the International Food Policy Research Institute (IFPRI). All analyses in this study adhered to the terms, conditions, and data usage guidelines set by IFPRI. Since the data is publicly accessible and de-identified (web links: https://doi.org/10.7910/DVN/OR6MHT, https://doi.org/10.7910/DVN/BXSYEL, https://doi.org/10.7910/DVN/NXKLZJ, this study did not require additional ethical review.

## Results

### Exploratory analysis

Table 3 summarizes descriptive statistics for continuous predictors of household dietary diversity. The average age of household heads was 43.47 years in 2012, increasing significantly over time, with the overall mean age reaching 45.55 years. Household size also grew across the rounds. Monthly non-food expenditure per capita decreased by about 40 Taka in 2015 compared to 2012 but rose to 355 Taka by 2018. In contrast, weekly food expenditure per capita increased significantly across the rounds, with an overall mean of 298 Taka. The mean area of cultivable land or pond operated by households was 73.5 decimals in 2012. Although it increased in 2015, it declined by 2.8 decimals in 2018.

Table 4 provides descriptive statistics for categorical variables, showing key trends. Female headship increased significantly over the rounds, with male-headed households decreasing from 83.11% in 2012 to 79.58% in 2018. The observed rise in female-headed households may be driven by factors such as men migrating for employment, evolving marital patterns, or changing societal norms that support women's empowerment. Exploring the underlying causes of this trend and its impact on household nutrition, especially for children, is a worthwhile area for research.

The percentage of households practicing Islam remained stable at around 89.35% across the study period. Educational attainment among household heads showed positive and significant progress. In 2012, 49.09% of household heads had no formal education, decreasing to 45.83% in 2015 and further to 45.23% in 2018. The proportion of household heads with secondary or higher education increased, while primary-level education rose in 2015 but

**Table 3. Descriptive Statistics of the Continuous Explanatory Variables.**

| Variable | Mean (Standard Deviation) | | | | Mean Difference (*t* value) | | |
|---|---|---|---|---|---|---|---|
| | Pooled | Round 1 (2012) | Round 2 (2015) | Round 3 (2018) | Round (1 vs 2) | Round (2 vs 3) | Round (1 vs 3) |
| Age of House | 45.55 | 43.47 | 45.61 | 47.56 | 2.14*** | 1.95*** | 4.09*** |
| hold Head (years) | (13.40) | (13.54) | (13.37) | (12.98) | (18.99) | (14.99) | (25.98) |
| Household Size | 4.62 | 4.05 | 4.61 | 5.21 | 0.57*** | 0.59*** | 1.16*** |
| | (1.75) | (1.44) | (1.64) | (1.92) | (39.43) | (39.88) | (51.23) |
| Nonfood Expenditure | 329.11 | 336.72 | 295.43 | 355.19 | -41.29*** | 59.76*** | 18.47* |
| Per Capita (1mo) | (357.46) | (501.38) | (243.51) | (266.08) | (-4.11) | (9.50) | (1.84) |
| Food Expenditure | 298.70 | 271.20 | 301.32 | 323.57 | 30.13*** | 22.24*** | 52.37*** |
| Per Capita (7d) | (191.50) | (164.02) | (209.07) | (195.00) | (7.18) | (5.25) | (13.37) |
| Cultivable Land/ | 75.86 | 73.51 | 78.44 | 75.64 | 4.93*** | -2.8* | 2.13 |
| Pond (decimal) | (128.76) | (126.30) | (139.62) | (119.51) | (3.44) | (-1.74) | (1.64) |

Notes: ***, **, and * indicate significance levels of 1%, 5%, and 10%, respectively.

**Table 4. Descriptive Statistics of the Categorical Explanatory Variables.**

| Variable | n (%) | | | | Adj. F value (p-value) | | |
|---|---|---|---|---|---|---|---|
| | Pooled | Round 1 (2012) | Round 2 (2015) | Round 3 (2018) | Round (1 vs 2) | Round (2 vs 3) | Round (1 vs 3) |
| *Characteristics of household head:* | | | | | | | |
| **Sex** | | | | | | | |
| Female | 2,557 (18.56%) | 776 (16.89%) | 844 (18.37%) | 938 (20.42%) | 10 48 | 14.71 | 34.72 |
| Male | 11,222 (81.44%) | 3,817 (83.11%) | 3,749 (81.63%) | 3,655 (79.58%) | (0.001) | (<0.001) | (<0.001) |
| **Religion** | | | | | | | |
| Islam | 12,311 (89.35%) | 4,112 (89.53%) | 4,101 (89.28%) | 4,099 (89.24%) | 3.70 | 0.47 | 4.15 |
| Others | 1,468 (10.65%) | 481 (10.47%) | 492 (10.72%) | 494 (10.76%) | (0.055) | (0.495) | (0.043) |
| **Education Level** | | | | | | | |
| Never Attended school | 6,437 (46.72%) | 2,255 (49.09%) | 2,105 (45.83%) | 2,078 (45.23%) | 28.15 | 5.98 | 34.64 |
| Primary or Lower | 3,472 (25.20%) | 1,104 (24.04%) | 1,197 (26.06%) | 1,171 (25.51%) | (<0.001) | (0.003) | (<0.001) |
| Secondary or Higher | 3,870 (28.08%) | 1,234 (26.87%) | 1,291 (28.11%) | 1,344 (29.26%) | | | |
| **Occupation** | | | | | | | |
| Farming | 5,329 (38.67%) | 1,935 (42.14%) | 1,770 (38.55%) | 1,623 (35.34%) | 17.85 | 16.30 | 62.40 |
| Others | 8,450 (61.33%) | 2,658 (57.86 %) | 2,823 (61.45%) | 2,970 (64.66%) | (<0.001) | (<0.001) | (<0.001) |
| **Marital Status** | | | | | | | |
| Married | 12,495 (90.68%) | 4,237 (92.25%) | 4,165 (90.68%) | 4,093 (89.12%) | 20.71 | 17.30 | 54.14 |
| Others | 1,284 (9.32%) | 356 (7.75%) | 428 (9.32%) | 500 (10.88%) | (<0.001) | (<0.001) | (<0.001) |
| *Other Household Characteristics:* | | | | | | | |
| **House Ownership** | | | | | | | |
| Owned | 13,242 (96.10%) | 4,343 (94.56%) | 4,430 (96.46%) | 4,469 (97.29%) | 13.28 | 5.06 | 26.41 |
| Others | 537 (3.90%) | 250 (5.44%) | 163 (3.54%) | 124 (2.71%) | (<0.001) | (0.025) | (<0.001) |
| **Access to Electricity** | | | | | | | |
| No | 4,870 (35.34%) | 2,375 (51.71%) | 1,857 (40.43%) | 638 (13.89%) | 76.52 | 249.40 | 395.72 |
| Yes | 8,909 (64.66%) | 2,218 (48.29%) | 2,736 (59.57%) | 3,955 (86.11%) | (<0.001) | (<0.001) | (<0.001) |
| **Access to Water** | | | | | | | |
| No | 1,595 (11.58%) | 1,178 (25.65%) | 417 (9.08%) | – | 192.03 | 115.09 | 422.47 |
| Yes | 12,184 (88.42%) | 3,415 (74.35%) | 4,176 (90.92%) | 4,593 (100.00%) | (<0.001) | (<0.001) | (<0.001) |
| **Negative Shock** | | | | | | | |
| No | 7,068 (51.29%) | 2,226 (48.45%) | 2,739 (59.63%) | 2,103 (45.79%) | 48.76 | 80.73 | 2.05 |
| Yes | 6,711 (48.71%) | 2,367 (51.55%) | 1,854 (40.37%) | 2,490 (54.21%) | (<0.001) | (<0.001) | (0.153) |
| **Total** | 13779 | 4593 | 4593 | 4593 | | | |

declined slightly by 2018. Farming as the primary occupation saw a decreasing trend, with an overall percentage of 38.67%. The proportion of married household heads also declined, though, on average, 90.68% of household heads were married across the study. Household ownership increased over time, with 96.10% of households owning their houses. Access to water and electricity improved significantly between 2012 and 2018. In 2012, only 48.29% of households had access to electricity, but this rose to 86.11% by 2018. Water access increased from 74% in 2012 to 100% in 2018. Negative shocks experienced by households decreased significantly from 2012 to 2015 but increased again in 2018.

Table 5 presents an overview of the outcome variables, highlighting their variation across different survey rounds. Household dietary diversity in rural Bangladesh shows a significant upward trend over time. In the first round (2012), the average Household Dietary Diversity Score (HDDS) was approximately 9.07, indicating that households consumed, on average, 9.07 out of 12 predefined food groups in a 7-day recall period. This average increased progressively in the subsequent rounds, reaching 10.10 by the third round in 2018. The overall mean HDDS across all rounds is 9.70, demonstrating a statistically significant increase in dietary diversity over time. Similarly, the Food Variety Score (FVS) showed a positive trend throughout the study. In 2012, the average household consumed 28.49 distinct or mixed foods during a 7-day recall period. By 2018, this value had increased significantly to 37.46. Notably, the largest increase occurred between 2012 and 2015. Although the change from 2015 to 2018 appears smaller, it remains statistically significant.

In rural areas, cereals, vegetables, miscellaneous foods, oils, and roots/tubers are consumed by nearly all households at least once a week (see Table 6). However, meat consumption isnotably lower, with only about 40% of households reporting meat intake within a seven-day

**Table 5. Descriptive Statistics of the Outcome Variables.**

| Variable | Mean (Standard Deviation) | | | | Mean difference (*t* value) | | |
|---|---|---|---|---|---|---|---|
| | Pooled | Round 1 (2012) | Round 2 (2015) | Round 3 (2018) | Round (1 vs 2) | Round (2 vs 3) | Round (1 vs 3) |
| HDDS | 9.70 (1.70) | 9.07 (1.75) | 9.94 (1.59) | 10.10 (1.58) | 0.87*** (23.47) | 0.15*** (4.34) | 1.03*** (26.67) |
| FVS | 33.36 (10.12) | 28.49 (8.40) | 34.13 (9.71) | 37.46 (10.05) | 5.65*** (21.25) | 3.33*** (10.84) | 8.97*** (29.76) |

Notes: ***, **, and * indicate significance levels of 1%, 5%, and 10%, respectively.

**Table 6. Percentage of Households Consuming Different Food Groups.**

| Food groups | Percentage (%) | | |
|---|---|---|---|
| | Round 1 (2012) | Round 2 (2015) | Round 3 (2018) |
| Cereals,rice,wheat | 100 | 99.85 | 99.73 |
| Roots and tubers | 96.98 | 97.81 | 98.03 |
| Vegetables | 99.98 | 99.84 | 99.71 |
| Fruits | 60.55 | 75.01 | 81.13 |
| Meat | 40.28 | 50.12 | 56.16 |
| Eggs | 57.51 | 73.64 | 76.23 |
| Fish | 93.81 | 94.52 | 94.76 |
| Pulses | 58.01 | 77.81 | 79.36 |
| Milk and milk products | 36.72 | 51.54 | 50.38 |
| Oil/Fats | 99.90 | 99.62 | 99.70 |
| Sugar/Honey | 63.73 | 74.73 | 74.50 |
| Miscellaneous | 99.62 | 99.91 | 99.92 |

period in the round 1. This indicates that almost 60% of households did not consume meat at least once a week. Encouragingly, meat consumption has risen by the third round of data collection, suggesting a positive trend. Similarly, dairy products were consumed by less than 40% of households at least once a week during the initial round, but their consumption has also risen in the following rounds. These trends indicate gradual improvements in the diversity of certain food items in rural diets over time.

From Fig 1, we can observe that the pattern is approximately similar across each division for both FVS and HDDS, with an increasing trend for each division over the years. Although Rangpur initially had the lowest Household Dietary Diversity Score (HDDS) among all the divisions, the data shows a noticeable improvement over time. Chittagong and Sylhet exhibit a better situation for dietary diversity compared to the other divisions.

## Results of the Poisson regression model

The fixed-effects Poisson regression model with robust standard errors was fitted with HDDS as the outcome variable. The results, presented in Table 7, reveal that education plays a crucial role in dietary diversity. Households with heads having primary or lower education show a 2.11% increase in the HDDS rate, while those with secondary or higher education exhibit a 2.05% increase compared to households where the head has no formal education. Farming as the primary occupation is associated with a 1.4% increase in HDDS. Furthermore, households with married heads experience a 2.9% higher HDDS rate than those with other marital statuses. The results show no statistically significant evidence that the household head's age or gender have influence on the HDDS. Households practicing Islam have a 5.51% lower HDDS rate compared to those following other religions, with this association significant at the 10% level.

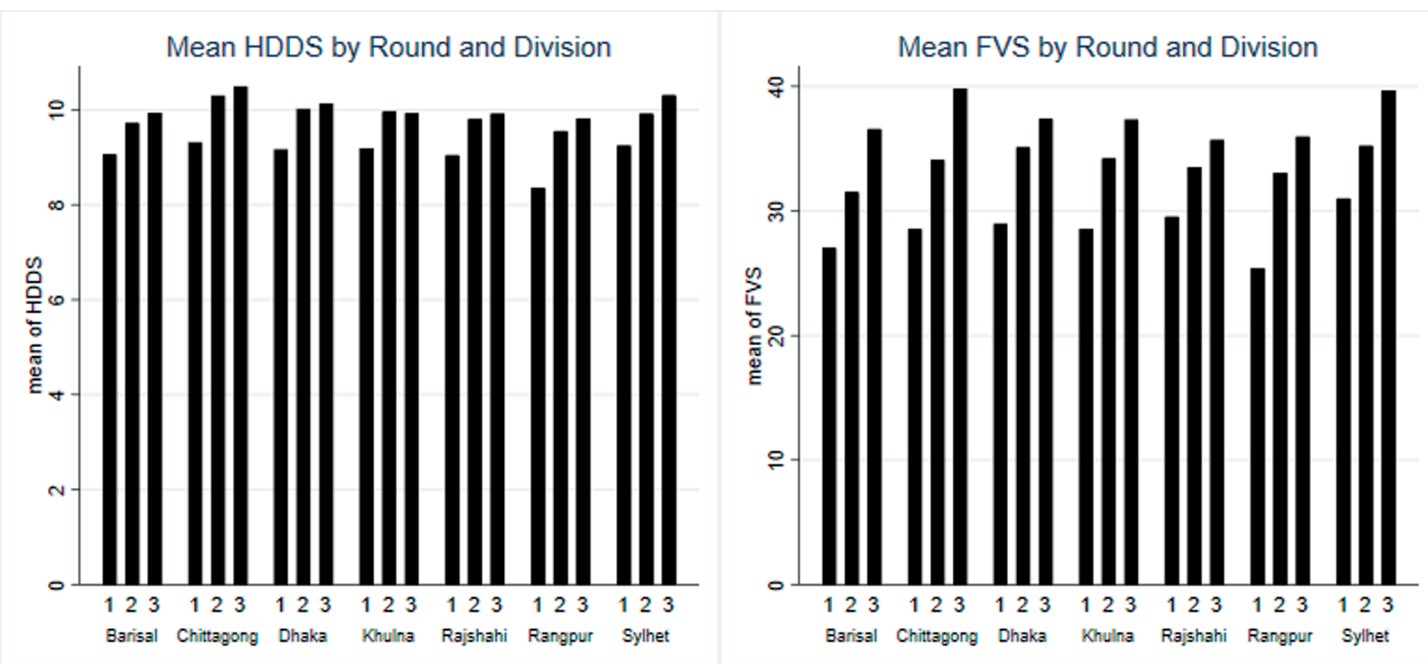

**Fig 1. Division and Round wise mean HDDS and mean FVS.**

**Table 7. Results of the Poisson Regression Model With HDDS as Outcome Variable.**

| Characteristic | IRR | (95 % CI of IRR) | p-value |
|---|---|---|---|
| *Characteristics of Household Head:* | | | |
| **Age** | 0.9999 | (0.9994, 1.0001) | 0.745 |
| **Sex** | | | |
| Female | – | – | |
| Male | 1.0007 | (0.9887, 1.0128) | 0.915 |
| **Religion** | | | |
| Islam | – | – | |
| Others | 1.0583 | (0.9985, 1.1216) | 0.056 |
| **Education Level** | | | |
| Never Attended School | – | – | |
| Primary or Lower | 1.0211 | (1.0056, 1.0368) | 0.008 |
| Secondary or Higher | 1.0205 | (1.0023, 1.0391) | 0.027 |
| **Occupation** | | | |
| Farming | – | – | |
| Others | 0.9863 | (0.9786, 0.9942) | 0.001 |
| **Marital Status** | | | |
| Married | – | – | |
| Others | 0.9714 | (0.9553, 0.9878) | 0.001 |
| *Others Household Characteristics:* | | | |
| **Household Size** | 1.0180 | (1.0143, 1.0217) | <0.001 |
| Cultivable Land or Pond | 1.0001 | (1.0000, 1.0001) | 0.002 |
| **Household Ownership** | | | |
| Owned | – | – | |
| Others | 1.0069 | (0.9909, 1.0231) | 0.401 |
| **Nonfood Expenditure Per Capita** | 1.0001 | (1.0000, 1.0000) | <0.001 |
| **Food Expenditure Per Capita** | 1.0002 | (1.0002, 1.0003) | <0.001 |
| **Access to Electricity** | | | |
| No | – | – | |
| Yes | 1.0096 | (1.0008, 1.0184) | 0.033 |
| **Access to Water** | | | |
| No | – | – | |
| Yes | 1.0185 | (1.0082, 1.0289) | <0.001 |
| **Negative Shocks** | | | |
| No | – | – | |
| Yes | 0.9984 | (0.9928, 1.0041) | 0.586 |
| **Round** | | | |
| 1 | – | – | |
| 2 | 1.0708 | (1.0641, 1.0777) | <0.001 |
| 3 | 1.0679 | (1.0585, 1.0774) | <0.001 |

Notes: CI = Confidence Interval, IRR = Incidence Rate Ratio

Household characteristics also influence dietary diversity. A one-person increase in household size corresponds to a 1.8% increase in the HDDS rate. Cultivable land, as well as food and non-food expenditures per capita, are positively associated with HDDS. Access to electricity and water also has significant positive effects, with incidence rate ratios of 1.0096 and 1.0185, respectively. Negative shocks do not significantly predict HDDS, whereas the survey rounds show significant differences. Round 2 and round 3 have more than a 7% higher HDDS rate compared to round 1, underscoring improvements over time.

Table 8 presents the analysis of the Food Variety Score (FVS) using a fixed-effects Poisson regression model with robust standard errors, highlighting key associations.

**Table 8. Results of the Poisson Regression Model With FVS as Outcome Variable.**

| Characteristic | IRR | (95 % CI of IRR) | p-value |
|---|---|---|---|
| *Characteristics of Household Head:* | | | |
| **Age** | 0.9996 | (0.9988, 1.0003) | 0.231 |
| **Sex** | | | |
| Female | – | – | |
| Male | 1.0390 | (1.0200, 1.0583) | <0.001 |
| **Religion** | | | |
| Islam | – | – | |
| Others | 1.0177 | (0.9335, 1.1095) | 0.691 |
| **Education Level** | | | |
| Never Attended School | – | – | |
| Primary or Lower | 1.0210 | (0.9993, 1.0432) | 0.058 |
| Secondary or Higher | 1.0114 | (0.9851, 1.0383) | 0.399 |
| **Occupation** | | | |
| Farming | – | – | |
| Others | 0.9787 | (0.9675, 0.9901) | <0.001 |
| **Marital Status** | | | |
| Married | – | – | |
| Others | 0.9820 | (0.9587, 1.0057) | 0.136 |
| *Others Household Characteristics:* | | | |
| **Household Size** | 1.0555 | (1.0496, 1.0615) | <0.001 |
| **Cultivable Land or Pond** | 1.0002 | (1.0001, 1.0002) | <0.001 |
| **Household Ownership** | | | |
| Others | 0.9852 | (0.9638, 1.0071) | 0.184 |
| Owned | – | – | |
| **Nonfood Expenditure Per Capita** | 1.0001 | (1.0000, 1.0000) | 0.006 |
| **Food Expenditure Per Capita** | 1.0006 | (1.0006, 1.0007) | <0.001 |
| **Access to Electricity** | | | |
| No | – | – | |
| Yes | 1.0049 | (0.9929, 1.0170) | 0.424 |
| **Access to Water** | | | |
| No | – | – | |
| Yes | 1.0355 | (1.0210, 1.0501) | <0.001 |
| **Shocks** | | | |
| No | – | – | |
| Yes | 0.9958 | (0.9877, 1.0040) | 0.313 |
| **Round** | | | |
| 1 | – | – | |
| 2 | 1.1255 | (1.1152, 1.1360) | <0.001 |
| 3 | 1.1803 | (1.1648, 1.1961) | <0.001 |

Notes: CI = Confidence Interval, IRR = Incidence Rate Ratio

Farming as the primary occupation of the household head is significantly associated with a 2.17% increase in the average FVS rate compared to other occupations. Additionally, household heads with primary-level education show a 4% higher FVS rate than those with no formal education, with this association being statistically significant at the 10% level. The gender of the household head is also a significant predictor, with households headed by males exhibiting a 4% higher FVS rate compared to those headed by females. This difference may be due to greater financial resources or decision-making power in male-headed households, allowing for more diverse food choices. In contrast, female-headed households may face economic constraints that limit food variety.

Several household characteristics are positively and significantly linked to FVS. For instance, a 1-unit increase in household size corresponds to a 5.6% increase in the average FVS rate. Similarly, greater cultivable land or pond area, higher food and non-food expenditures, and access to water are all positively associated with FVS. Specifically, access to water increases the FVS rate by 3.5%. Survey rounds further reveal notable differences in FVS. Round 2 shows a 12% higher FVS rate compared to round 1, while round 3 exhibits an 18% increase compared to round 1. In contrast, negative shocks, household ownership, and secondary or higher education of the household head are not found to be significant predictors of FVS, as shown in Table 8.

## Discussion

This study aimed to explore the socioeconomic and demographic factors influencing dietary diversity in rural Bangladesh, utilizing longitudinal data from three rounds of the Bangladesh Integrated Household Survey (BIHS). Our findings provide valuable insights into how various factors affect household dietary diversity, measured through the Household Dietary Diversity Score (HDDS) and the Food Variety Score (FVS). Using fixed-effects Poisson regression allowed us to account for the count nature of the data and control for unobserved heterogeneity across households. The analysis identified an upward trend in dietary diversity across survey rounds, underscoring the evolving food consumption patterns in rural Bangladesh. The results highlight that dietary diversity is significantly associated with multiple socioeconomic factors, including the education and occupation of the household head, household size, and expenditure patterns. These findings emphasize the role of both individual characteristics and broader household dynamics in shaping dietary quality.

**The role of education.** Among the socioeconomic determinants that emerged from our analysis, the education level of the household head is a significant predictor of dietary diversity. Households headed by individuals with primary education showed a positive association with both HDDS and FVS at varying significance levels. This finding aligns with existing literature, which consistently highlights education as a crucial factor influencing food choices and dietary practices [29,30]. Education of household head enhances decision-making, enabling investments in children's education to break the poverty cycle and improve food security, nutrition, and socioeconomic status. Food literacy also helps households make informed, affordable choices, improving dietary diversity. While education plays a crucial role in enhancing dietary diversity, approximately 45% of household heads still had no formal education in 2018 (round 3). Although this figure has been steadily decreasing from round 1 to round 3 of the survey, it remains significant and concerning. The lack of education among household heads, particularly in rural areas, poses a considerable challenge, as these individuals are responsible for making major decisions that affect their families. To address this issue, implementing educational programs—that are at least comparable to primary education—for older adults in rural areas could significantly improve dietary diversity scores. Such programs would enhance their knowledge of nutrition, caloric intake, and informed decision-making regarding food choices.

**The impact of occupation and agriculture.** Occupation also plays a vital role in determining dietary diversity. Our findings indicate that households with farming as the primary occupation exhibited higher dietary diversity. A number of factors, such as economic capacity and accessibility to diverse dietary sources, may contribute to this association. However, the primary reason is that the household head's direct exposure to locally farmed foods significantly enhances the family's dietary options, especially when farming is the main occupation. For instance, rural farming households often cultivate various crops and raise livestock,

thereby increasing access to a broader range of food categories, which can directly enhance dietary diversity. Despite a decline in the proportion of households engaged in farming—from 42.14% in 2012 to 35.34% in 2018—agriculture remains a crucial livelihood source in rural Bangladesh. This suggests that while farming provides access to a variety of food sources, there is a need for policies that support farmers through training in sustainable practices and diversification strategies. Agricultural diversification has been shown to enhance dietary diversity by increasing the variety of foods produced and consumed [11]. Therefore, targeted interventions aimed at improving farming techniques and access to resources could significantly impact dietary quality in these communities.

**Gender, household size, and expenditure in food security.** The gender of household head is a significant predictor, with male-headed households showing a 4% higher FVS. This difference may be due to greater financial resources or decision-making power in male-headed households, allowing for more diverse food choices. In contrast, female-headed households may face economic constraints that limit food variety. Household size and per capita food and non-food expenditures also emerged as critical determinants of dietary diversity. The number of household members is positively associated with dietary diversity, as households with more members often benefit from shared resources and collective purchasing power. Conversely, lower food expenditure per capita is associated with reduced dietary diversity, reflecting the constraints faced by rural households. This aligns with previous research demonstrating that financial limitations significantly affect dietary choices, leading to less varied diets and increasing the risk of malnutrition [31]. Though a family can access less expensive, diverse foods, but certain food items like meat and milk products are costly and require greater financial spending to include in a regular diet. Analysis of the data shows that on average, rural households consume more than 9 out of 12 food categories weekly, indicating diversity in their diets. However, this diversity does not extend to all the food groups necessary for optimal nutrition, with consumption of meat and milk products being notably lower. This suggests that rural families are generally avoiding more expensive food choices despite having access to diverse options. Moreover, longitudinal trends show that as food expenditure increases over time, the average weekly consumption of meat also increases. This demonstrates a clear trade-off between financial spending and food choices—households prioritize their financial resources strategically, investing more in costly food options like meat when their income or expenditures allow. This finding reflects the economic constraints rural households face and their dietary prioritization behavior, highlighting the complexity of food choice decisions under limited resources. As such, improving economic conditions and expanding access to financial resources for rural households could enhance their ability to purchase diverse food items.

**Negative shock, religion, marital status, and time.** The significance of negative shocks and marital status on dietary diversity is not stable in this study. Marital status can influence dietary diversity indirectly through factors such as household size, food expenditure, or the availability of home-grown food, which may vary depending on the circumstances. When these mediating factors are accounted for in the model, they likely explain much of the variation attributed to marital status, making its direct effect on dietary diversity less significant. Negative shocks, such as natural disasters, job losses, or illnesses, are generally expected to reduce dietary diversity by decreasing household food availability and expenditure. Variables like household ownership, cultivable land, or food expenditure might mediate the relationship between negative shocks and dietary diversity. Since these factors are included in the model, the direct impact of negative shocks may become insignificant, as their effects are captured indirectly. Negative shocks could have different levels of impact depending on household resilience (e.g., savings, alternative income sources). Some households may prioritize dietary

diversity despite economic constraints, while others may experience significant reductions, leading to inconsistent findings. In the context of Bangladesh, where the majority of the population is Muslim, religion is an important cultural factor that could influence dietary practices. Our findings show that individuals following Islam exhibit lower dietary diversity in both models. One potential reason for this negative association could be linked to cultural dietary practices common among Muslim households, such as fasting or certain dietary restrictions. However, it is important to note that our sample is not balanced in terms of religion, and the findings suggest that religion is not a statistically significant determinant of dietary diversity in this context. Therefore, while the observed negative effect might align with fasting or other cultural practices, we cannot confidently make direct comparisons due to these limitations. Time itself is shown to be positively and significantly associated with both variables of dietary diversity. Advancements in agricultural techniques, such as the introduction of hybrid seeds and improved rice varieties, have contributed to better food availability and improved agricultural output. These technological improvements have likely influenced food intake patterns in rural households and can be partially explained by changes across the survey rounds. The increasing access to mobile phones between 2012 and 2019 provided rural populations in Bangladesh with greater integration into financial and agricultural markets. Programs leveraging mobile platforms, such as e-agriculture initiatives and mobile banking systems, have been instrumental in enhancing food security and dietary diversity [32].

**Strengths and limitations of this study.** This study's use of fixed-effects Poisson regression provides robust insights by accounting for unobserved household-level heterogeneity. However, the fixed-effects model cannot capture time-invariant factors, such as geographical location, which may also influence dietary diversity. Another limitation is the use of 7-day recall data instead of the more widely used 24-hour recall method. The 7-day recall was chosen due to practical constraints, as the dataset from the BIHS relied on this format for dietary data collection. However, the 7-day recall serves as a valid alternative, offering a more comprehensive view of a household's dietary intake, minimizing the risk of capturing data from an atypical day [19,33]. While the 7-day recall method may introduce potential biases, such as inaccuracies in memory recall, it also offers certain advantages in rural settings. For instance, it captures variations in dietary patterns over a longer period, which can provide a more comprehensive view of food consumption in households with limited market access or fluctuating food availability. This broader time frame allows for a better representation of dietary patterns, including variations in food availability and consumption across the week. Another limitation is the use of HDDS, as excluding foods consumed outside the home may lead to an incomplete assessment of dietary diversity, particularly in settings where such consumption is more prevalent. The study is also limited by the potential for endogeneity and selection bias, as the fixed-effects model may not fully account for these challenges. Future research could explore advanced modeling approaches, such as instrumental variables or propensity score matching, to address these limitations. Finally, the study's reliance on secondary data imposes inherent limitations related to data availability and quality.

This study offers several key strengths that enhance the reliability and relevance of its findings. First, the use of a three-round balanced panel dataset from the BIHS allows for robust longitudinal analysis, capturing changes in household dietary diversity over time and controlling for unobserved heterogeneity through fixed-effects Poisson regression. Second, by employing both the HDDS and FVS as outcome variables, the study provides a detailed understanding of dietary diversity from multiple perspectives. Third, additional robustness checks, including random-effects models and quadratic terms for expenditures, are presented in Appendix A, confirming the stability of our key findings. Furthermore, the focus on rural Bangladesh, an underrepresented area in nutrition research, contributes valuable

insights into the socioeconomic determinants of dietary practices in this context. Finally, the study's alignment with the Sustainable Development Goals (SDGs), particularly Goal 2 (Zero Hunger), and Goal 3 (Health and Well-being), enhances its potential impact and policy relevance.

## Conclusion

This study utilized balanced panel data from three rounds of the BIHS to explore the socioeconomic and demographic determinants of dietary diversity in rural Bangladesh, using HDDS and FVS as outcome measures. The analysis identified several significant factors influencing dietary diversity, including household size, farming as the main occupation, access to cultivable land or ponds, water access, food and non-food expenditures, and the primary education level of the household head.

Rural households not only prioritize food expenditures but also allocate financial resources to non-food items necessary for their livelihood. Items such as agricultural tools, transportation, healthcare, or education expenses often take precedence because they are essential for maintaining income sources and ensuring family stability. This means that households frequently face trade-offs between spending on food diversity and meeting other essential non-food needs. These trade-offs reflect economic constraints and strategic decision-making as families attempt to balance their limited resources across multiple priorities. As food expenditures increase, households may opt to spend more on meat consumption. However, this choice comes at the cost of reducing spending on other food categories or even essential non-food items. These decisions highlight how rural families must carefully allocate their budgets, often sacrificing immediate food diversity in favor of long-term sustainability or livelihood security. The education of the household head enhances awareness of nutrition and cost-effective choices, promoting a diverse diet. Agricultural practices, such as crop cultivation and livestock rearing, provide fresh, homegrown foods and reduce market reliance, improving dietary diversity. Financial resources enable access to varied, nutrient-rich foods and support investments in agriculture and storage, ensuring food security. Addressing education, agriculture, and financial stability together is key to improving dietary diversity in rural settings. Our findings highlight that improving dietary diversity is essential for addressing malnutrition and food insecurity in rural areas. Although dietary diversity has improved over time, the persistence of household heads lacking formal education (45.23% in 2018) underscores the need for adult education programs to further enhance food security. Farming continues to play a critical role in both dietary diversity and rural food security, calling for policies that promote sustainable agricultural practices and support for farmers.

In conclusion, policymakers should focus on promoting education, supporting sustainable agriculture, and expanding access to financial resources to ensure households can afford diverse diets. Strengthening these areas will contribute to achieving food security, reducing malnutrition, and improving the overall quality of life in rural communities.

# Appendix

## Appendix A. Robustness checks

In this study, we analyzed two dietary diversity indicators: the Household Dietary Diversity Score (HDDS) and the Food Variety Score (FVS), with the latter serving as an additional measure of robustness. Our initial analysis revealed that several factors—education level, occupation, household size, cultivable land, access to water, food and non-food expenditure, and time—were consistently significant predictors for both HDDS and FVS, with similar directionality and significance levels in the fixed-effects model. However, it is important to note that the fixed-effects model does not account for time-invariant variables, such as geographical location.

To address this limitation, we estimated a random-effects model for both HDDS and FVS (see Table 9), incorporating division as a geographical location variable, with Barisal serving as the base category among the seven divisions: Barisal, Chittagong, Dhaka, Khulna, Rajshahi, Rangpur, and Sylhet. The random-effects model revealed that division significantly influenced dietary diversity, with Chittagong, Rajshahi, Khulna, and Dhaka showing a positive

**Table 9. Results (IRRs) of Poisson Regression for Fixed-Effect and Random-Effect Models, with Standard Errors in Parentheses.**

| Model | Fixed effect | Random effect | Fixed effect | Random effect |
|---|---|---|---|---|
| **Variables** | **HDDS** | | **FVS** | |
| **Age** | 0.9999 | 0.9998* | 0.9996 | 0.9991*** |
| | (0.0003) | (0.0001) | (0.0004) | ( 0.0002 ) |
| **Religion** | | | | |
| Islam | – | – | – | – |
| Others | 1.0583* | 1.0111** | 1.0177 | 1.0622*** |
| | (0.0314) | ( 0.0043) | ( 0.0449 ) | (0.0079) |
| **Sex** | | | | |
| Female | – | – | – | – |
| Male | 1.0007 | 0.9935 | 1.0390*** | 1.0455*** |
| | (0.0061) | (0.0041) | (0.0098) | (0.0065 ) |
| **Education Level** | | | | |
| Never Attended School | – | – | – | – |
| Primary or Lower | 1.0211*** | 1.0326*** | 1.0210* | 1.0424*** |
| | (0.0080) | (0.0037) | (0.0112) | (0.0057 ) |
| Secondary or Higher | 1.0205** | 1.0581*** | 1.0114 | 1.0685 |
| | (0.0094) | (0.0038) | (0.0136) | (0.0061) |
| **Occupation** | | | | |
| Farming | – | – | – | – |
| Others | 0.9863*** | 0.9743*** | 0.9787*** | 0.9725*** |
| | (0.0040) | (0.0029) | (0.0058) | (0.0045) |
| **Access to Electricity** | | | | |
| No | – | – | – | – |
| Yes | 1.0096** | 1.0359*** | 1.0049 | 1.0325*** |
| | (0.0045) | (0.0034) | (0.0061) | (0.0049) |
| **Household Ownership** | | | | |
| Owned | – | – | – | – |
| Others | 1.0069 | 0.9981 | 0.9852 | 0.9806** |
| | (0.0082) | (0.0068) | (0.0111) | (0.0090) |
| **Marital Status** | | | | |
| Married | – | – | – | – |
| Others | 0.9714*** | 0.9632*** | 0.9820 | 0.9648*** |
| | (0.0082) | (0.0057) | ( 0.0120) | (0.0083) |

*(Continued)*

**Table 9.** (Continued).

| Model | Fixed effect | Random effect | Fixed effect | Random effect |
|---|---|---|---|---|
| **Household Size** | 1.0180*** | 1.0186*** | 1.0555*** | 1.0531*** |
| | (0.0019) | (0.0009) | (0.0030 ) | (0.0016) |
| **Cultivable Land or Pond** | 1.0001*** | 1.0001*** | 1.0002*** | 1.0002*** |
| | (<0.0001) | (<0.0001) | (<0.0001) | (<0.0001) |
| **Food Expenditure (Per Capita)** | 1.0002*** | 1.0003*** | 1.0006*** | 1.0007*** |
| | (<0.0001) | (<0.0001) | (<0.0001) | (<0.0001) |
| **Nonfood Expenditure (Per Capita)** | 1.0000*** | 1.0000*** | 1.0000*** | 1.0000*** |
| | (<0.0001) | (<0.0001) | (<0.0001) | (<0.0001) |
| **Access to Water** | | | | |
| No | – | – | – | – |
| Yes | 1.0185*** | 1.0349*** | 1.0356 | 1.0475*** |
| | (0.0053) | (0.0047) | (0.0074) | (0.0067) |
| **Negative Shock** No | – | – | – | – |
| Yes | 0.9985 | 0.9943** | 0.9958 | 0.9911 |
| | (0.0029) | (0.0025) | (0.0041) | (0.0037) |
| **Round** | | | | |
| 1 | – | – | – | – |
| 2 | 1.0708*** | 1.0624*** | 1.1255*** | 1.1206*** |
| | (0.0035) | (0.0031) | (0.0053) | (0.0048) |
| 3 | 1.0679*** | 1.0500*** | 1.1803*** | 1.1687*** |
| | (0.0048) | (0.0037) | (0.0080) | ( 0.0064 ) |
| **Division** | | | | |
| Barisal | – | – | – | – |
| Chittagong | - | 1.0187*** | - | 1.0234** |
| | | (0.0063) | | (0.0105) |
| Dhaka | - | 1.0156*** | - | 1.0640*** |
| | | (0.0058) | | (0.0101) |
| Khulna | - | 1.0149** | - | 1.0762*** |
| | | (0.0069) | | (0.0122) |
| Rajshahi | - | 1.0130* | - | 1.0720*** |
| | | (0.0070) | | (0.0117) |
| Rangpur | - | 0.9985 | - | 1.0635*** |
| | | (0.0076) | | (0.0128) |
| Sylhet | - | 0.9969 | - | 1.0287*** |
| | | (0.0066) | | (0.0114 ) |

Notes: ***, **, and * indicate significance levels of 1%, 5%, and 10%, respectively.

and significant association relative to Barisal for both HDDS and FVS. We had also considered several potential two-way interactions, including sex and education level, sex and marital status, marital status and house ownership, and education and occupation, to explore their potential effects on dietary diversity. In the fixed-effects model, none of these interaction terms were significant for either the HDDS or the FVS. However, the random-effects model revealed a significant negative association for some of the interactions (e.g. education and occupation). Nevertheless, given that the Hausman test supports the fixed-effects model for this study, the results from the random-effects model may not be reliable. As a result, we proceeded with the analysis without including these interaction terms.

To ensure that our regression models did not oversimplify the dynamics of dietary diversity, we re-estimated the models with a quadratic term for both food and non-food expenditures (see Table 10), recognizing that expenditures might follow a diminishing return pattern. Even with this adjustment, the coefficients' magnitude, direction, and significance for key predictors (education level, occupation, household size, cultivable land, access to water, food and

**Table 10. Results (IRRs) of Fixed-Effect Poisson Regression Model with Quadratic Terms, with Standard Errors in Parentheses.**

| Variables | HDDS | FVS |
|---|---|---|
| **Age** | 1.0000 | 0.9996 |
| | (0.0003) | (0.0004) |
| **Religion** | | |
| Islam | – | – |
| Others | 1.0593* | 1.0207 |
| | (0.0309) | ( 0.0454 ) |
| **Sex** | | |
| Female | – | – |
| Male | 0.9954 | 1.0292** |
| | (0.0061) | (0.0095) |
| **Education Level** | | |
| Never Attended School | – | – |
| Primary or Lower | 1.0207** | 1.0202* |
| | (0.0079) | (0.01212) |
| Secondary or Higher | 1.0217** | 1.0143 |
| | (0.0093) | (0.0140) |
| **Occupation** | | |
| Farming | – | – |
| Others | 0.9852*** | 0.9764*** |
| | (0.0039) | (0.0055) |
| **Access to Electricity** | | |
| No | – | – |
| Yes | 1.0094** | 1.0053 |
| | (0.0044) | (0.00597) |
| **Household Ownership** | | |
| Owned | – | – |
| Others | 1.0068 | 0.9847 |
| | (0.0081) | (0.0106) |
| **Marital Status** | | |
| Married | – | – |
| Others | 0.9745*** | 0.9874 |
| | (0.0082) | ( 0.0115) |
| **Household Size** | 1.0199*** | 1.0591*** |
| | (0.0018) | (0.0028 ) |
| **Cultivable Land or Pond** | 1.0001*** | 1.00015*** |
| | (.000022) | (.00003) |
| **Food Expenditure (Per Capita)** | 1.0005*** | 1.0011*** |
| | (0.000063) | ( .0001109 ) |
| **Food Expenditure (Per Capita) Squared** | 0.9999*** | 0.9999*** |
| | (5.72e-08) | ( 9.92e-08 ) |
| **Nonfood Expenditure (Per Capita)** | 1.00002*** | 1.000004 |
| | (5.40e-06) | (8.22e-06) |
| **Nonfood Expenditure (Per Capita) Squared** | 1.0000** | 1.0000 |
| | (4.40e-10) | (6.32e-10) |
| **Access to Water** | | |
| No | – | – |
| Yes | 1.0175*** | 1.0332*** |
| | (0.0052) | (0.00716) |
| **Negative Shock** | | |
| No | – | – |
| Yes | 0.9994 | 0.9982 |
| | (0.0028) | (0.0039) |
| Round | | |
| 1 | – | – |
| 2 | 1.0687*** | 1.1207*** |
| | (0.0034) | (0.00513) |
| 3 | 1.0604*** | 1.16415*** |
| | (0.0047) | (0.0075) |

Notes: ***, **, and * indicate significance levels of 1%, 5%, and 10%, respectively.

non-food expenditure, and time) remained stable across both HDDS and FVS. This stability underscores the robustness of our findings: these factors consistently exhibit a positive association with dietary diversity, regardless of the model specification or dietary diversity indicator used.

## Author contributions

**Conceptualization:** Tayaba Cheragee Prachee, Md Rasel Biswas.

**Data curation:** Tayaba Cheragee Prachee.

**Formal analysis:** Tayaba Cheragee Prachee, Md Rasel Biswas, Saiful Islam.

**Investigation:** Md Rasel Biswas.

**Methodology:** Tayaba Cheragee Prachee, Md Rasel Biswas.

**Supervision:** Md Rasel Biswas, Saiful Islam.

**Validation:** Saiful Islam.

**Writing – original draft:** Tayaba Cheragee Prachee.

**Writing – review & editing:** Md Rasel Biswas.

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
