## [Decision Letter · Decision Letter 0]

12 Nov 2024

PONE-D-24-47837 Exploring the socioeconomic determinants of dietary diversity in rural bangladesh: A longitudinal study PLOS ONE

Dear Dr. Biswas,

Thank you for submitting your manuscript to PLOS ONE. After careful consideration, we feel that it has merit but does not fully meet PLOS ONE’s publication criteria as it currently stands. Therefore, we invite you to submit a revised version of the manuscript that addresses the points raised during the review process.

Kindly address all the comments given by the reviewer 1 and 2

We look forward to receiving your revised manuscript.

Kind regards,

George N Chidimbah Munthali

Academic Editor

PLOS ONE

Journal Requirements:

Reviewers' comments:

Reviewer's Responses to Questions

**Comments to the Author**

1. Is the manuscript technically sound, and do the data support the conclusions?

Reviewer #1: Yes

Reviewer #2: Yes

2. Has the statistical analysis been performed appropriately and rigorously? 

Reviewer #1: Yes

Reviewer #2: Yes

3. Have the authors made all data underlying the findings in their manuscript fully available?

Reviewer #1: Yes

Reviewer #2: Yes

4. Is the manuscript presented in an intelligible fashion and written in standard English?

Reviewer #1: Yes

Reviewer #2: Yes

5. Review Comments to the Author

Reviewer #1: Figures and tables: The tables are well labelled and easy to understand. Review Table 4 and I would start with this table (demographics) then move to table 3 – just a suggestion.

Methods: The methods are sufficient.

Study design: Sampling – A little more detail to the construction of the balanced panel dataset to allow for replicability of arriving at the 4,593 per year (line 71 -81).

Statistics: The statistical tests used (Poisson regression model) covered good ground and are appropriate for count data. Overdispersion of count data was taken care of.

Results and data: The results are presented very well though it would be nice to have the results cover the demographics first (line 178 - 190).

The female headship increased – in the discussion this should be pointed out for potential follow-up studies (line 201).

Gender of household head – it would be nice to have a discussion around this result too (line 243 -244)

The paper might need to describe what “Larger households” means in their context (line 296).

Limitations of the methods could also mention working with secondary data (line 307)

Conclusion: The conclusion could be improved (line 327 – 336).

References: 1996 reference might be a little old, is it possible to use a more recent citation? (line 2-3).

Reviewer #2: This manuscript comprehensively explores the socioeconomic and demographic factors influencing dietary diversity in rural Bangladesh. Below is my review comments:

Abstract:

The abstract provides a clear and concise summary of the study, effectively outlining its primary objective, methodology, key findings, and potential implications. The study aims to explore the socioeconomic and demographic factors influencing dietary diversity in rural Bangladesh, using longitudinal data from three rounds of the Bangladesh Integrated Household Survey (BIHS). The use of fixed-effects Poisson regression is appropriate for accounting for unobserved heterogeneity across households, and the study employs both the Household Dietary Diversity Score (HDDS) and the Food Variety Score (FVS) as measures of dietary diversity. The results reveal an upward trend in dietary diversity over time and highlight the significant role of socioeconomic factors, such as the education level and occupation of the household head, household size, and expenditure patterns, in shaping dietary diversity. These findings underscore the importance of both individual characteristics and broader household dynamics in influencing dietary quality. The abstract also emphasizes the policy relevance of the findings, suggesting the need for educational programs to enhance knowledge of nutrition and food choices, as well as interventions to support sustainable agricultural practices and improve economic conditions for rural households. While the abstract is well-structured, it could benefit from greater specificity regarding the magnitude of the effects of key variables and a clearer explanation of the theoretical framework guiding the study. Also, placing the findings within a broader global context, particularly in relation to the Sustainable Development Goals (SDGs) on food security, would further highlight the significance of the study’s contributions to both academic and policy discussions on improving dietary diversity and food security in rural communities.

Introduction

The introduction of this study offers a solid foundation for understanding dietary diversity and its relevance to food security in rural Bangladesh. The paper appropriately defines two key indicators—Household Dietary Diversity Score (HDDS) and Food Variety Score (FVS)—which are central to the study’s analysis. These indicators are well-established in the literature and are effectively framed as tools to assess both the nutritional adequacy and the access to diverse foods within households. The review of existing studies adds value by providing a broader context and showcasing the varied factors, such as food expenditure, farm production diversity, and market access, which influence dietary diversity. This review is particularly relevant as it identifies gaps in the existing literature, notably the lack of longitudinal studies in rural Bangladesh and the limitations of cross-sectional data in understanding temporal dynamics.

However, while the introduction highlights significant socioeconomic determinants of dietary diversity, it could benefit from a clearer identification of specific research gaps in rural Bangladesh, particularly with respect to factors such as gender, access to healthcare, and regional disparities. The introduction also introduces the Poisson regression model as a methodological approach but does not provide sufficient detail on why this model is particularly appropriate for the study, which could enhance the reader’s understanding of the analytical rigor. Additionally, while the study mentions the use of panel data, a more explicit discussion of how this approach contributes to the robustness of the findings would be valuable. Overall, the introduction presents a well-structured argument for the study but could improve by elaborating on the specific contributions of the research to the existing body of knowledge.

Data, Sampling and Variables

The study includes various independent variables such as age, sex, education, occupation of the household head, household size, and food expenditures. These socioeconomic factors are essential for understanding dietary diversity and food security in low-income settings. However, while the study covers many critical factors, it could benefit from a deeper focus on gender, especially in terms of intra-household food distribution and decision-making, as gender dynamics often play a significant role in food security.

The operationalization of HDDS and FVS is generally clear, though excluding food consumed outside the home in the HDDS calculation may limit the comprehensiveness of dietary diversity assessment. Additionally, variables such as access to markets, local agricultural policies, or seasonal variations in food availability are not considered, which could be significant in rural settings. Despite these limitations, the study provides a valuable analysis of the socioeconomic factors influencing dietary diversity in rural Bangladesh. It offers recommendations for further research, particularly in incorporating gender and regional factors.

Exploratory Analysis

The study employs sound analytical methods to examine household dietary diversity in rural Bangladesh, using Poisson regression to model the count data outcome variables, Household Dietary Diversity Score (HDDS) and Food Variety Score (FVS). Poisson regression is appropriate for count data, as it assumes that the dependent variable follows a Poisson distribution, and the study accounts for overdispersion by using a robust estimator and pseudo-likelihood functions. This approach ensures that unobserved heterogeneity and intra-household correlation are properly addressed, enhancing the reliability of the findings.

The exploratory analysis is well-conducted, with descriptive statistics and adjusted F-statistics to examine the distribution of continuous and categorical variables across survey rounds. The use of adjusted F-statistics is particularly noteworthy, as it accounts for survey design features such as clustering, stratification, and unequal weighting, which is critical in a complex survey design. Additionally, the study uses a Hausman test to determine the most appropriate model, finding that the fixed-effects Poisson regression model is the best fit, consistent with previous research.

However, while the methodology is robust, some potential limitations need consideration. First, while the fixed-effects model controls for unobserved household-specific heterogeneity, it may not fully address temporal effects or interactions between unobserved variables over time. The study also assumes linear relationships, which may oversimplify the dynamics of dietary diversity. Further, the study could benefit from exploring alternative models to handle potential endogeneity or selection bias, particularly given the socio-economic variables involved.

Results

The manuscript utilizes Poisson regression to model the count data outcome variables, Household Dietary Diversity Score (HDDS) and Food Variety Score (FVS), which is an appropriate methodological choice given the nature of the data. Poisson regression is well-suited for count data and allows for the modelling of overdispersion through robust estimators and pseudo-likelihood functions. This methodological approach ensures that unobserved heterogeneity is accounted for, enhancing the reliability of the study's findings. However, I recommend expanding the rationale behind the choice of Poisson regression, particularly for a broader audience, by providing a more detailed explanation of its advantages and why it is the most suitable method for this type of analysis.

The manuscript also examines several socioeconomic factors influencing dietary diversity, but there is a need for a more in-depth exploration of why certain variables, such as marital status and negative shocks, show inconsistent associations across the two dietary diversity indices (HDDS and FVS). A deeper understanding of these inconsistencies could provide valuable insights into how these factors interact with other variables or influence dietary diversity in different contexts.

Additionally, the findings related to religion, specifically the potential influence of cultural or dietary practices among Muslim households, should be further contextualized. Given the significant role that religion can play in shaping dietary habits, especially in rural settings, it would be beneficial to discuss how specific practices, such as fasting or dietary restrictions, may impact dietary diversity in Muslim households.

The manuscript would also benefit from a more detailed discussion of potential confounders or omitted variables, such as income or wealth, which could influence the results. The omission of these factors could bias the findings and affect the interpretation of the study’s implications for policy. Including a consideration of these variables would enhance the robustness of the analysis and provide a more nuanced understanding of the socioeconomic determinants of dietary diversity.

Lastly, while the manuscript reports improvements in dietary diversity over time, it would be helpful to clarify how the survey rounds themselves may have influenced these results. For instance, changes in household circumstances, regional interventions, or broader socio-economic shifts could contribute to the observed improvements. A discussion on how these temporal factors interact with the study’s findings would strengthen the manuscript and provide additional context for interpreting the changes in dietary diversity across the survey rounds.

Discussions

The use of longitudinal data is a key strength, enabling the authors to capture changes in dietary diversity over time, which is a significant advantage over cross-sectional studies that provide only a snapshot of data. The application of fixed-effects Poisson regression is appropriate, given the count nature of the outcome variables (HDDS and FVS), and the methodology effectively controls for unobserved heterogeneity across households, enhancing the internal validity of the study. This approach is robust and helps mitigate the potential confounding effects of omitted variables. The dual measures of dietary diversity—HDDS and FVS—add depth to the analysis, providing a nuanced understanding of dietary patterns that strengthens the reliability of the findings.

The study’s alignment with the Sustainable Development Goals (SDGs), particularly those addressing food security and nutrition, is commendable, and the findings offer valuable insights for policymakers aiming to improve rural nutrition. Further, the focus on rural Bangladesh, an underrepresented context in nutrition research, is an important contribution to the literature. However, several areas could be further strengthened:

1. Addressing Time-Invariant Factors: While the use of a fixed-effects model is appropriate for controlling unobserved heterogeneity, the inability of this model to account for time-invariant factors, such as geographical location, is a limitation. The authors briefly mention this, but further discussion on how these unmeasured factors may influence dietary diversity would enhance the interpretation of the results. Exploring alternative methodologies, such as random-effects models, could also provide additional insights into the role of these factors.

2. Mechanisms Behind Education's Role: The paper identifies education, particularly primary education, as a key determinant of dietary diversity. However, the study could benefit from a more detailed exploration of the mechanisms through which education influences dietary practices. A deeper discussion of how education impacts food choices, such as through increased nutritional awareness or improved economic decision-making, would provide a clearer understanding of the pathways linking education to dietary diversity.

3. Farming and Dietary Diversity: The relationship between farming as the primary occupation and greater dietary diversity is intriguing but requires further exploration. It would be helpful to investigate the underlying mechanisms driving this association. For example, is the increased dietary diversity a result of farmers having access to a wider variety of food sources, or is it due to higher income levels that enable more diverse food purchases? A more detailed analysis of agricultural practices and economic outcomes would help clarify these pathways.

4. Expenditure Patterns: The study identifies food and non-food expenditures as important determinants of dietary diversity, but further discussion is needed on how these expenditures translate into food choices. Are higher food expenditures associated with a preference for more expensive, diverse foods, or do households prioritize bulk purchases of cheaper, less diverse options? Understanding how households allocate their financial resources in the context of dietary choices would enhance the depth of the analysis.

5. Methodological Transparency: While the authors mention the use of 7-day recall data, a more thorough explanation of why this method was chosen over the 24-hour recall would improve the clarity and transparency of the methodology. Also, addressing potential biases associated with recall-based dietary assessments, particularly in rural settings where memory recall may be less reliable, would strengthen the validity of the results.

6. Interactions Between Determinants: The study identifies several socioeconomic determinants of dietary diversity, but it would benefit from a deeper analysis of how these factors interact with each other. For instance, how does the education level of the household head interact with occupation or household size in influencing dietary diversity? Understanding these interactions could provide a more comprehensive picture of the complex dynamics influencing dietary diversity.

Conclusion

The conclusion of the manuscript provides a solid summary of the key findings and offers valuable insights into the determinants of dietary diversity in rural Bangladesh, with clear implications for policy. The authors effectively emphasize the need for improvements in education, support for sustainable agriculture, and expanded financial resources to enhance dietary diversity. These recommendations are timely and relevant. However, the conclusion could be strengthened by further elaborating on the underlying mechanisms that link the identified determinants to dietary diversity. Specifically, a more detailed discussion of how education, agricultural practices, and financial resources influence dietary choices would provide a deeper understanding of the dynamics at play. Additionally, the conclusion would benefit from a more explicit consideration of time-invariant factors, such as geographical location, and how these may influence the results. A deeper exploration of the role of financial constraints, particularly in terms of how households allocate their resources, would also enhance the study’s practical applicability. Overall, while the conclusion offers important contributions to understanding the socioeconomic factors affecting dietary diversity in rural settings, incorporating these additional elements would further strengthen the manuscript’s robustness and policy relevance.

6. PLOS authors have the option to publish the peer review history of their article (what does this mean?). If published, this will include your full peer review and any attached files.

Reviewer #1: No

Reviewer #2: No

---

## [Author Response · Author response to Decision Letter 1]

29 Dec 2024

We have attached a letter that responds to each point raised by the reviewers. Thanks.

---

## [Editor Report · Decision Letter 1]

2 Jan 2025

Exploring the socioeconomic determinants of dietary diversity in rural Bangladesh: A longitudinal study

PONE-D-24-47837R1

Dear Dr.  Md Rasel Biswas,

We’re pleased to inform you that your manuscript has been judged scientifically suitable for publication and will be formally accepted for publication once it meets all outstanding technical requirements.

Kind regards,

George N Chidimbah Munthali

Academic Editor

PLOS ONE
---

## [Editor Report · Acceptance letter]

PONE-D-24-47837R1

PLOS ONE

Dear Dr. Biswas,

I'm pleased to inform you that your manuscript has been deemed suitable for publication in PLOS ONE. Congratulations! Your manuscript is now being handed over to our production team.

Kind regards,

on behalf of

Mr George N Chidimbah Munthali

Academic Editor

PLOS ONE